# Metallocavitins as Advanced Enzyme Mimics and Promising Chemical Catalysts

Albert A. Shteinman

Department of Kinetics and Catalysis, Institute of Problems of Chemical Physics of RAN, Prospect of Academic Semenov, 1, 142432 Chernogolovka, Russia; as237t@icp.ac.ru

**Abstract:** The supramolecular approach is becoming increasingly dominant in biomimetics and chemical catalysis due to the expansion of the enzyme active center idea, which now includes binding cavities (hydrophobic pockets), channels and canals for transporting substrates and products. For a long time, the mimetic strategy was mainly focused on the first coordination sphere of the metal ion. Understanding that a highly organized cavity-like enzymatic pocket plays a key role in the sophisticated functionality of enzymes and that the activity and selectivity of natural metalloenzymes are due to the effects of the second coordination sphere, created by the protein framework, opens up new perspectives in biomimetic chemistry and catalysis. There are two main goals of mimicking enzymatic catalysis: (1) scientific curiosity to gain insight into the mysterious nature of enzymes, and (2) practical tasks of mankind: to learn from nature and adopt from its many years of evolutionary experience. Understanding the chemistry within the enzyme nanocavity (confinement effect) requires the use of relatively simple model systems. The performance of the transition metal catalyst increases due to its retention in molecular nanocontainers (cavitins). Given the greater potential of chemical synthesis, it is hoped that these promising bioinspired catalysts will achieve catalytic efficiency and selectivity comparable to and even superior to the creations of nature. Now it is obvious that the cavity structure of molecular nanocontainers and the real possibility of modifying their cavities provide unlimited possibilities for simulating the active centers of metalloenzymes. This review will focus on how chemical reactivity is controlled in a well-defined cavitin nanospace. The author also intends to discuss advanced metal–cavitin catalysts related to the study of the main stages of artificial photosynthesis, including energy transfer and storage, water oxidation and proton reduction, as well as highlight the current challenges of activating small molecules, such as $H_2O$, $CO_2$, $N_2$, $O_2$, $H_2$, and $CH_4$.

**Keywords:** supramolecular chemistry; cavitins; biomimetics; metalloenzymes; metallocavitins; methane

## 1. Introduction

Biological catalysts–enzymes, usually demonstrate excellent selectivity and reactivity. Their mode of functioning is complex and far from completely understood. Metalloenzymes are ubiquitous and responsible for a wide range of challenging chemical transformations that proceed under mild conditions and with high chemo-, regio- and stereo-selectivity. Cavities and pores, being an integral feature of protein bodies in nature, are formed by folding and self-assembling of polypeptide helices through non-covalent and partially covalent interactions. In metalloenzymes they serve to accommodate the active sites for the delivery and bonding of substrates and the excretion of products (Figure 1a). The enzyme pocket–cavity in the protein body, plays a key role in the control of metal center nuclearity, substrate binding, substrate–catalyst–reactant pre-association, regio- and stereo-selectivity and substrate–product in/out exchanges [1]. It also provides a well-defined second co-ordination sphere for the activation and/or stabilization of intermediate reactive species and protects the metal center from undesired pathways. The local microenvironment

of the metal center in the pocket differs substantially from the bulk solution. Cavities around the active sites of enzymes are of low symmetry and contain different chemical functionalities, such as recognition sites, catalytic groups and conformational switches. Chiral discrimination is one of the fundamental processes in enzymes. Functions of the metal complexes in the cavity are strongly dependent on its properties, including accessible spin states, oxidation potential and Lewis acidity. These properties are further fine-tuned by a well-defined first and second coordination sphere (Figure 1b).

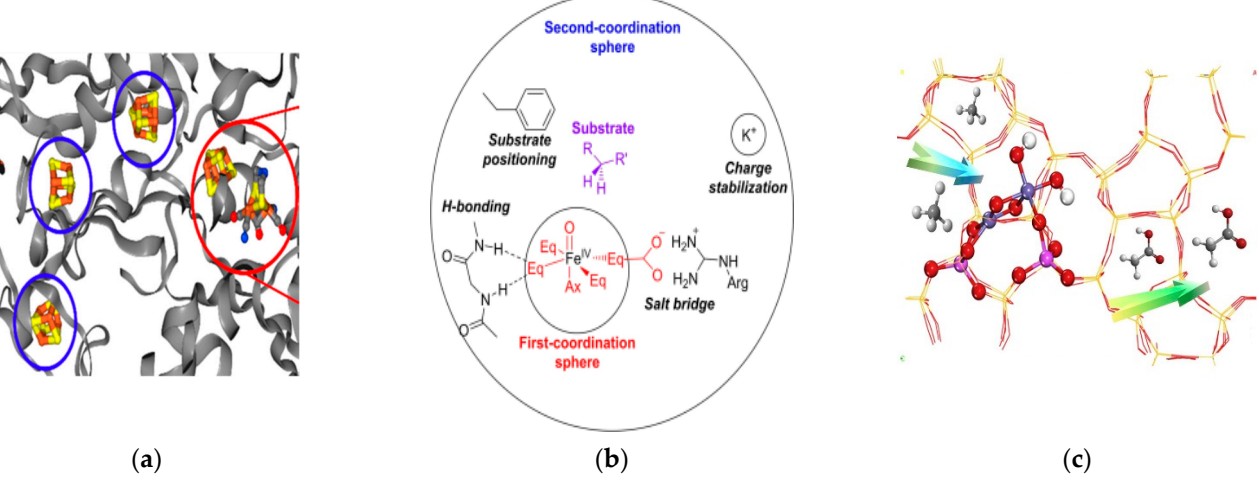

(**a**)  (**b**)  (**c**)

**Figure 1.** General performance mechanism of M-cavitin catalysts. (**a**) Crystal structure of [FeFe] hydrogenase (*Clostridium pasteurianum* CpI; PDB: 4XDC), showing [FeFe]-S(Cys)-[4Fe4S] active clusters included in its cavities. Adapted with permission from (Castner et al., 2021) [109]. Copyright 2021 American Chemical Society. (**b**) Cavity effects. Reprinted with permission from (Mukherjee et al., 2021) [29]. Copyright 2021 American Chemical Society. (**c**) Conversion of methane to $CH_3COOH$ on [$Fe^{III}$-$(\mu O)_2$-$Fe^{III}$]-ZSM-5, the arrows show the transport of substrates in and products out. Reprinted with permission from (Wu et al., 2022) [72]. Copyright 2023 Elsevier.

The former is directly involved in metal coordination and usually consists of mixtures of different donor functionalities. In contrast, the second coordination sphere is not directly involved in metal binding but connected with the first by weak and reversible non-covalent interactions, such as hydrogen-bonding, electrostatic interactions, acid–base chemistry, van der Waals and hydrophobic forces. The second coordination sphere regulates the catalytic processes, proton or electron shuttling and substrate–product transport and determines the activity and selectivity of metalloenzymes. The pursuit of broadening our fundamental understanding of enzymatic catalysis has inspired scientists to develop and explore smaller synthetic complexes as enzyme mimics [2]. There are two main aims of mimicking enzyme catalysis: (1) curiosity, to gain insight into the nature of enzyme active sites and (2) practical tasks of mankind, to learn from nature and adopt from her long evolution experience. For a long time the traditional metalloenzyme modeling was mainly focused on the first coordination sphere of the metal ion and the second coordination sphere could be introduced directly only via the related chelate ligands of the first coordination sphere. In traditional homogeneous catalysis activity, the selectivity and stability of a transition metal catalyst was controlled by the ligands of the first coordination sphere. In biomimetic catalysis, a new direction for the research of advanced cavity-like models of metalloenzymes has gradually appeared and formed [3]. Many important aspects of enzymatic chemistry have been investigated by supramolecular chemistry, including molecular self-assembly, folding, molecular recognition, and host–guest chemistry on classical macrocyclic hosts provided by synthetic organic chemistry, such as cyclodextrins, crown ethers, cyclophanes, and calixarenes [4]. These gave rise to covalent cavitins [4,5]. On the other hand, supramolecular chemistry, built on weak and reversible non-covalent interactions, has emerged as a pow-

erful and versatile strategy for the fabrication of coordination cavitins and has led to the creation of molecular containers or cages [5,6] and porous polymer molecules such as the metal-organic frameworks (MOF) and covalent-organic frameworks (COF) (Figure 2) [7,8].

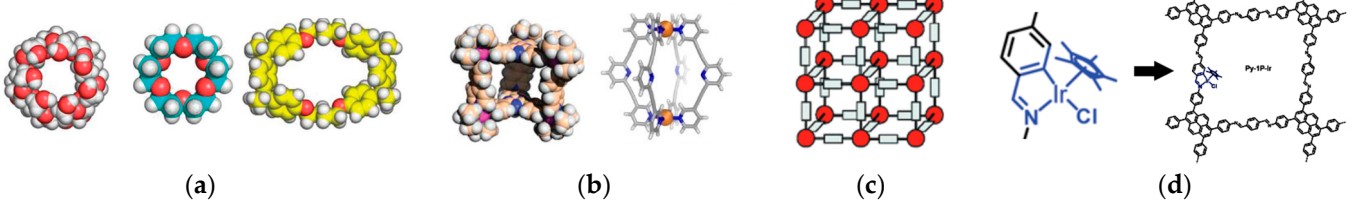

**Figure 2.** Diversity of cavitins. Discrete cavitins: (**a**) cyclodextrin, crown ether, cyclophane, and (**b**) metal-coordination cages. Extended cavitins: (c) MOF and (**d**) COF, the arrow symbolizes the incorporation of the M-complex into the COF cavity (Hu, J et al., 2021) [23].

The extended cavitins quickly gained recognition as industrial heterogeneous catalysis due to their outstanding characteristics: high-porosity and high-density catalytic metal centers, remarkable sorption properties, shape selectivity and easy syntheses in preparative quantities [9]. Metal catalysts can be encapsulated in various types of cavitins, providing the tools to control their activity and selectivity via the second coordination sphere. The supramolecular strategy has become more and more dominating in biomimetic chemistry over the last decade [10,11]. Due to their cavity-like structure and convenient modification, cavitins provide unlimited possibilities to mimic the active sites of natural enzymes.

## 2. The Diversity of Cavitins

Microporous compounds, such as charcoal or zeolites [12], have been used for a long time as carriers for metal ions in heterogeneous catalysts because they greatly increase the performance of the encapsulated transition metal. Cavity macrocycles, such as cyclodextrins, crown ethers, cyclophanes and calixarenes, have been studied as host molecules in the field of molecular recognition, which is key for the high catalytic efficiency and selectivity of natural enzymes [13]. To mimic the cavity and pores of natural enzymes, a number of polycycle molecular containers based on covalent bonding, such as carcerands, hemicarcerands, cryptophans, capsules and cages, have been prepared during the last decade. Among the classes of covalent cavitins the most popular are a derivatives of the cyclotriveratrylene (resorcinarene) [14] as well as cavitins formed by bonding resorcinarene units [15]. In the ocean of covalent and non-covalent cavitins there are two big classes: discrete individual molecule monocavitins (Figure 2a,b), such as cyclodextrins, calixarenes, cryptands, cucurbiturils, metal–organic cages (MOC), covalent organic cages (COC), helicates [16], and many others; and extended ones, polycavitins (Figure 2c,d), such as zeolites, MOF, COF, porous polymers, porous molecular crystals [17], hollow [18] and dynamic [19] MOF, and others which all together demonstrate a rich library of architectures varying in shape, size and geometry. H-bonded capsules based on resorcinarene units (Figure 3a) and H-organic frameworks (HOF) are evolving into novel and important classes of cavitins. Metal-COF (MCOF) are also emerging as a bridge between MOFs and COFs via integrating metal active sites into COFs (see Figure 2d) [20]. In recent years, there has been a growing interest in more exotic classes of cavitins, such as porous liquids and metal foams. A porous liquid (PL) is a liquid that combines the cavity of porous solids with the fluidity of liquids. The permanent pores endow PL unique physicochemical properties, interesting for catalysis, particularly for photocatalysis. A metal foam is a cellular structure consisting of a solid metal with pores comprising a large portion of its volume. They are considered as promising catalyst carriers due to their high porosity, large specific surface area, and satisfactory thermal and mechanical stability [21]. It is generally assumed that monocavins are more suitable for the modeling and academic study of enzyme active sites, while polycavitins are used for the fabrication of advanced heterogeneous catalysts. However, in the

recent years the appearance of MOCs in catalysis has also increased. Using self-assembly and incorporating different functional groups, complex supramolecular hosts with diverse shapes, sizes and chemical environments of the cavity have been easily designed from relatively simple components [15,21].

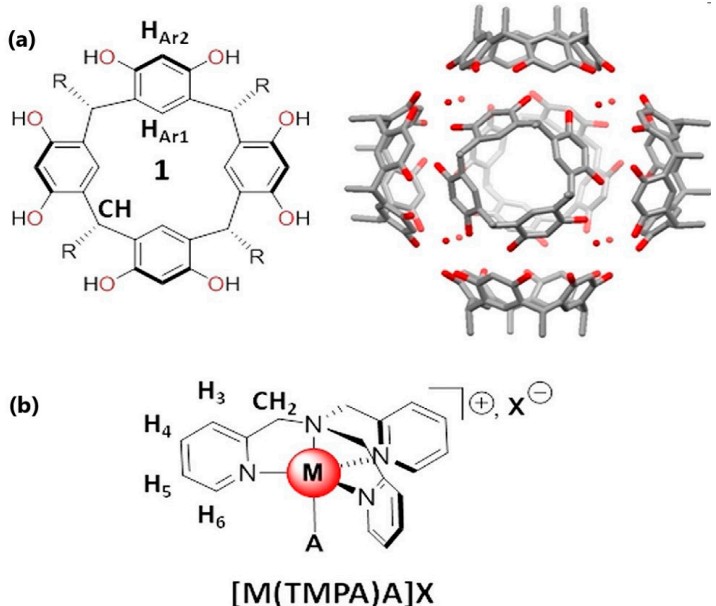

**Figure 3.** Encapsulation of a metallocomplex (**b**) in capsule (**a**). (**a**) The resorcin[4]arene **1** (R=$C_{11}H_{23}$) and structure of the H-bonded capsule. Reprinted with permission from (Zhang et al., 2021) [20]. Copyright 2021 Elsevier.

*2.1. Metallocavitins*

MOFs as MOCs are formed by coordination-driven self-assembly. They are composed of polydentate organic linkers and inorganic nodes containing metal ions or clusters known as secondary binding units (SBU). The metal in the coordination cavitins may serve not only in constructive goals but also as a coordinatively unsaturated catalytic center. Other approaches for the incorporation of metal active sites into cavitins include metallolinkers, non-covalent encapsulation of metal complexes and enzymes, templated metal–ligand assemblies and post synthetic metallation [3]. The most popular Schiff base COF possesses uniformly distributed imine linkages, which revealed a new metal binding mode. The imine linkage in COFs is the most tunable bond among all the currently employed reversible COF linkages and readily chelating transition metal via cyclometalation [22]. For example, the iridacycle B-decorated COF (Figure 2d) exhibited more than 10-fold efficiency than its molecular analog in photocatalytic hydrogen evolution from aqueous formate solution under mild conditions [23]. The robust porosity, stability, and chemical functionality of COF can be controlled by the reasonable selection of organic building blocks. Chirality is associated with the origin of life on Earth and plays a great role in the functioning of metalloenzymes. COFs are shown to be capable of inducing chiral molecular catalysts from non-enantioselective to highly enantioselective in organic reactions [24]. On the other hand, a method to synthesize chiral MOFs from achiral precursors by modifying the substituents utilizing chiral fragments was reported recently [25]. While the robust porosity of MOFs and COFs renders them as promising heterogeneous catalysts, they suffer from diffusion problems in mass transportation. Due to this, approach for the polymerization of soluble MOCs [26,27], and the preparation of semi-heterogeneous metal–enzyme-integrated catalysts using soluble porous imine molecular cages [23] were developed.

### 2.2. Design and Characterization

Design strategies have employed subcomponent self-assembly via the simultaneous formation of dynamic coordinative (N→metal), covalent (N=C), and other bonds. A key facet of metallo-supramolecular self-assembly is predicting the products of self-assembly based on constituent metal ion geometry and ligand conformation [28]. Binding selectivity, created by weak non-covalent interactions between the hosts and guests, is influenced by the size, shape and flexibility of cavitin [29]. A general design problem is that the linker units and SBUs should provide a unique flexibility/rigidity balance and directionality for the ligands to achieve the desired geometry and optimal host–guest interactions [30]. The right balance between the flexibility and rigidity of cavitin is favorable for binding substrates and releasing products [31]. Shape-persistent organic cavitins permit the precise control of their size, geometry, and the presence of functional groups in the interior of their cavities [32]. Adaptability is a hallmark of enzymes and flexible cavitins can mimics this via structural changes that accompany adsorption and desorption steps [33]. Host flexibility can greatly affect the cavity size and shape and lead to behaviors analogous to the induced fit of substrates within the active sites of enzymes. A little structural flexibility is inherent to some "rigid" metal–organic hosts, but torsional twisting of trigonal prismatic cages leads to a dramatic change in cavity size [34]. With advances in single-crystal X-ray diffraction and economic methods of computational structure optimization, cavity sizes can be readily determined. Practically very useful, simple rules, such as Rebek's 55% rule [35], fail to take into account structural flexibility that can allow hosts to significantly adapt their internal cavity [34]. Computational analysis offers a potential route to quantitatively examine the flexibility of metalorganic assemblies and may be used in the design of cavitins [36]. For example, a computational screening method able to predict new cavitins [37]. The Toolkit cgbind facilitates the characterization and prediction of functional metallocages [38]. A tight binding chemical method (GFN-xTB) has been developed specifically for geometry optimization in large molecular systems [39]. The volume calculations on empty cages and prospective guests with the online utility Voss Volume Voxelator confirmed that $Fe_4(Zn-L)_6$ has the appropriate size to accommodate a hydroformylation catalyst [39]. To explain the catalytic activity of the two dipalladium(II) cages the molecular dynamics simulation was explored for the evaluation of their conformational flexibility [38]. Hydrophobic MOFs have unique advantages as catalysts for various reactions: the hydrophobicity is beneficial for substrates to access the active sites and can improve the water stability of the MOF, they are also able to achieve spontaneous separation from the hydrophilic new products, thus improving selectivity [40]. Reversible bond formation is one of the prime prerequisites for the crystallization of cavitins. A general procedure to grow large single crystals of three-dimensional imine-based COFs was developed [41] using the principles of dynamic covalent chemistry [42,43]. In the design of complex cavitins it is necessary to take into account the balance between reversibility and robustness of the connecting bonds for enhanced crystal growth [44]. Many useful MOFs with enhanced catalytic performance possess varying degrees of chemical instability hampering their practical applications. The MOF/parylene-N hybrid not only imparts the chemical stability of an MOF without obviously impacting their inherent nature, but also broadens the scope of this catalysts in different aqueous environments [45]. A novel strategy for the synthesis of a highly crystalline and porous cyanurate-linked COF (CN-COF) by dynamic nucleophilic aromatic substitution was reported recently [46]. CN-COFs contain flexible backbones that exhibit unique AA′-stacking due to the interlayer H-bond interactions, exhibiting good stability. The complexity of cavitins has increased dramatically over the years. Heteroleptic, mixed-metal, hybrid and low symmetry assemblies are becoming more commonplace [47,48]. Improvements within SCXRD and the advancement of computational power allow the rapid and in-depth analysis of these systems [34]. Polyoxometalates (POMs) exhibit unique chemical properties that make them very attractive as catalysis. The ring-shaped lacunary POM comprises inorganic cavitins containing a large cavity useful for accumulating metal-cations. Recently an original approach was developed for the selective synthesis of

multinuclear Cu-containing ring-shaped POM Cu4-Cu16 by the stepwise addition of four of copper(II) acetate equivalents (Figure 4) [49]. Insertion of POM into an MOF opens up new opportunities in heterogeneous catalysis [50].

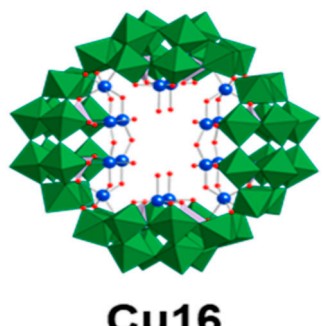

**Figure 4.** An example of an inorganic cavitin: multinuclear Cu-containing ring-shaped POM Cu16. Adapted with permission from (Koizumi et al., 2022) [49]. Copyright 2022 Wiley.

Perovkskites are another purely inorganic cavitin, which has recently received great attention as catalysts [51] due to their large surface area, low density, and high loading capacity. A typical perovskite oxide has the general formula $ABO_3$, in which A is a lanthanum or an alkaline earth metal and B is a transition metal. The enthusiasm for perovskite oxides is that they show a highly flexible elemental composition, with a large variation in properties that can be tailored by doping design. Progress in metallo-supramolecular chemistry has created the potential to synthesize metallocavitins with more than one function within the same assembly [47]. The inspiration has come from enzymes that congregate, for example, a substrate recognition site, an allosteric regulator element and a reaction center. The formation of heteroternary cucurbit[8]uril-viologen–naphthol complexes led to bifunctional photoredox catalysts for hydrogen generation [52]. Novel cubic cages with a different polarity to the peripheral environment surrounding the cage encapsulating catalytically active cobalt(II) meso-tetra(4-pyridyl) porphyrin were synthesized for study of polarity effects in cyclopropanation reactions [53]. The work [54] opens new perspectives for the synthesis of a more diverse library of coordination nanocages with innovative structures, metal ion composition and functionality.

### 3. Cavity Effects

Enzymes are a source of imagination and inspiration for chemists. They incorporate multiple functionalities in their substrate-binding cavities, in order to achieve high selectivity and activities. A good example of the cavity effect on a catalytically inactive binuclear iron complex was demonstrated in a paper [55] devoted to [FeFe]hydrogenase modeling. The complex $Fe_2[\mu-(SCH_2)_2NH](CN)_2(CO)_4^{2-}$ was shown to integrate into the inactive apo-form of [FeFe]-hydrogenases to yield a fully active enzyme. The cavities perform substrate encapsulation, molecular transformation, intermediate capturing, and product release, which facilitated the catalytic cycle. Cavity effects are based on entropy effects, cage-wall effects, absorption, desorption, shape- and size-selectivity [56] and include second sphere and hydrogen-bonding interactions, salt-bridges, and long-range allosteric effects from bound cations and anions (Figure 1b) and, at last, from component interactions. Basic atomic-molecular properties drastically change upon confinement within the catalyst framework, leading to effects such as increased excitation energy and lower polarizability, which can be explained using the "particle in the box" as a simplified model [57]. For larger molecules, spectroscopic evidence for the intrinsic decrease in the p-p* gap in aromatic hydrocarbons, such as naphthalene and anthracene, has been shown [57]. More recent studies, however, revealed that the pure effect of confinement rather leads to an increase in the HOMO–LUMO gap and point toward the specific properties of electrostatic stabilization within the active catalytic site [57]. The transition metal in the majority of

enzymes is the center of their activity, but the cavity itself can perform some activation of the substrate, isolation of metal complexes to prevent aggregation or decomposition and increase the local concentration of catalyst and reaction partners. For example, the intermediary-sized cucurbit[7]uril in aqueous solution selects and effectively accelerates (in $4 \times 10^5$) the endo dimerization of cyclopentadiene. DFT calculations suggest that catalysis is due to an entropy dominated transition-state stabilization in the tightly packed ternary reaction complex [58]. In another example, the rigid, spherical cavity in spherical cavitin quantitatively encapsulates azobenzene and stilbene derivatives with 100% cis-selectivity in water and their cis-azo isomerization is suppressed due to the confinement effect [59]. The Narazov cyclization, which needs an acidic media, in the metallocage $[Ga_4L_6]^{12-}$ can proceed at pH 8. In this case, an acceleration comparable to some enzymes is observed, which is caused by the preorganization of the encapsulated substrate and stabilization of the transition state. The experimental results and quantum chemical calculations reveal that a $Ga_4L_6{}^{12-}$ cage accelerates the cyclization reactions of pentadienyl alcohols because of an increase in the basicity of the complexed alcohol [60,61]. The design of new catalytically effective cavitin is limited because of our poor understanding of cavity effects in depth. A simple and effective DFT protocol was suggested, which takes into account both the thermodynamic and kinetic aspects of catalysis permitting the elucidation of many effects on the molecular level [36]. The protection of the reactive functional groups favors reaction at the unprotected sites. Cavity effects alter the typical reaction pathway, switching reactions on and off [62], operating substrate selection based on size and shape, leading to unusual selectivity or inducing stereoselectivity through asymmetric scaffolds. Most importantly for the reaction rate are the proximity and orientation of the substrates and transition-state stabilization. The concepts of 'confinement effect' and 'second sphere effects', often used in modeling enzymes, are not clear enough and are often unjustifiably substituted or identified. In the active center of the enzyme there are a number of effects that affect the catalyzed reaction: purely geometric, such as a limited space, size and shape of the cavity, physicochemical, associated with the microenvironment, chemical, such as covalent and non-covalent bonds of the 'second sphere' functional groups with the reaction participants, long-term effects such as electrostatic [63], allosteric and the effects of inter-component interactions with other enzymes. As a result of these effects the confined molecules can fundamentally change their chemical and physical properties compared to those in bulk solution. Cavity effects (CE) are directed to the preorganization of catalysts and reactants, and confinement effects, identified within the second sphere, affect the reaction itself. Significant progress in our understanding of CE in cavitins was reached due to single molecule fluorescence microscopy imaging (SMS) over the last decade [64]. This technique was developed to directly monitor the behaviors of individual molecules in a confined space, thus enabling spatial and temporal visualization and a better understanding of molecular dynamics. Additionally, it was also observed that confinement induced reactivity change, on and off switching of reactions, substrate selection, stereoselectivity, regioselectivity, and product distribution variation.

*3.1. Isolation from Bulky Solvent, Selective Incorporation and Stabilization of the M-Complex and Reactants*

The separation of the complex from the bulk solvent and the control of the in/out exchange is reminiscent of the roles of the protein backbone in metalloenzymes. The cavity can change the structure of the active site and prevent the catalyst from decomposing. The neutral complex Ru(II) (Figure 5) was encapsulated inside a self-assembled hexameric host similar to (a) (Figure 3, [20]). Different spectral data and molecular dynamics simulations support the inclusion and motions of the complex inside the capsule. The embedded complex was assessed by the $NaIO_4$ catalytic oxidation aryl-methyl alcohols into aldehydes, which is dependent on the substrates' size in the order benzyl > 4-phenyl-benzyl > 9-anthracenemethanol [65]. No discrimination between the substrates was observed in the absence of the cavitin.

**Figure 5.** Neutral ruthenium(II) complex as a catalyst of arylmethyl alcohol oxidation (Hkiri S. et al., 2022) [65].

Usually cavitins control the nuclearity of the M-complex. For example, the encapsulation of metal complex (b) in monomer form has been demonstrated in a dynamic H-bonded capsule (a) (Figure 3) [20]. The position of the M-complex inside this capsule can be derived from NMR analysis and confirmed by docking simulations [19]. Assembling phenothiazine into a cavitin improves its photocatalytic performance and stability due to less aggregation inducing its quenching and also due to preorganization of the electron donor–acceptor complex within the cage [66]. $Au_{25}$ nanocluster encapsulated into MOF loses surface ligands and exhibits superior activity and stability in the oxidative esterification of furfural [67]. Enzymes bind reactants within their pockets and reduce the distance between the M-center and the substrates. Inspired by enzymatic behavior, cavities can be engineered to co-encapsulate metal catalysts with substrates in such a way that after M-complex encapsulation, some space remains vacant for the co-encapsulation of substrates. Substrates of appropriate size and shape react with the confined M-complex due to the thermodynamically favorable host–guest binding process of molecular recognition based on complementary physicochemical characteristics, which allows acquisition and orientation. Substrate selectivity is difficult to rationalize for small molecules, such as $H_2$, $O_2$, $CO_2$, and $CH_4$, that possess a range of physical characteristics too narrow to allow either precise positioning or discrimination between the reactants. Nevertheless, metalloenzymes have evolved to metabolize these substrates with high selectivity and efficiency due to small-molecule tunnels and gate-effects, for example, for the selective oxidation of methane [68]. Substrate may adopt a high-energy conformation that is structurally similar to the transition state, thus leading to a lowered activation energy. Conformational changes during substrate binding frequently appear in enzymes. Cavitins help to investigate this effect. The preferred conformation and orientation of the bound guest determines the molecular behavior in the cavity. The conformation is dependent upon the intrinsic encapsulation capability of the hosts. For example, for long chain alkanes or fatty acids, bent binding motifs are often observed when they are sequestrated. This can result in the close proximity of terminal reactive functional groups through enforced orientation of the guest. Guest encapsulation induces an entropic penalty that is often compensated for by favorable entropic and enthalpic gains from desolvation of the guest and the liberation of high-energy solvent molecules from the binding site [56]. This also involves the shielding of specific reactive groups by supramolecular encapsulation. For example, the electrophilic $\alpha$-carbon on a $[PhN_2]^+$ ion can be selectively deactivated upon host–guest complexation with cucurbit[7]uril in aqueous media, achieving a 60-fold increase in the half-life of the carbocation. However, the electrophilic nitrogen of the encapsulated diazonium ion remains active towards diazo coupling with strong nucleophiles in water [69]. Electrostatic contributions are known as primary factors in enzyme catalysis. However, for a long time there were no models to study this mechanism. Positively charged hosts are able to attract negatively charged guests (and vice versa) and guest binding affinity, driven by electrostatic interactions, which can be modulated by different solvents [63]. Remarkable examples of reactive guest stabilization by confinement in synthetic molecular contain-

ers has been reported, involving the stabilization and detection of reaction intermediates through the formation of thermodynamically stable and covalent host–guest complexes with functionalized resorcin[4]arene cavitins [70]. A discrete nanocage of core−shell design with a hydrophilic interior and a hydrophobic exterior (Figure 6a) was able stabilize metal complexes (Figure 6b) with an uncommon oxidation state in organic solvents [71].

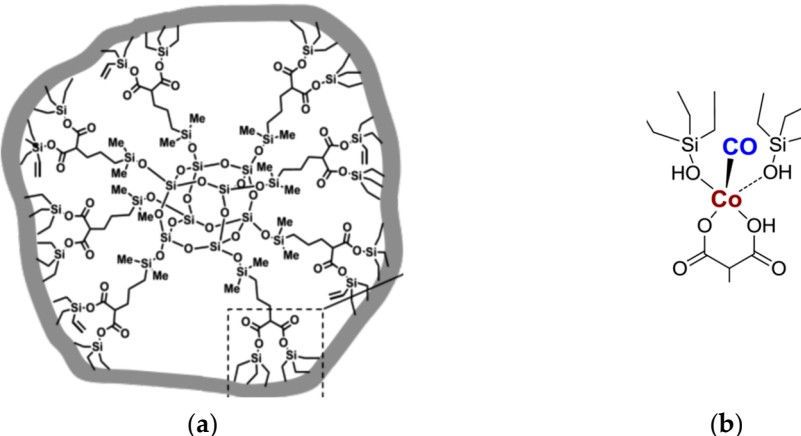

(a)　　　　　　　　　　　　　　　　(b)

**Figure 6.** Stabilizing complex (**b**) with Co(I) oxidation state in a discrete nanocage (**a**) of core−shell design (Shen et al., 2014) [71].

Steric groups in the second coordination sphere define a corridor for the approach of substrates into the active sites. For instance, iron picket-fence porphyrin complexes have bulky amide substituents positioned in a single facial orientation, thereby constructing a cavity for small molecules, such as dioxygen [29]. The cavity effects on the catalytic reaction dynamics under variable nanopore morphologies, including pore length and diameter at the single-molecule level, were studied and were found to be dependent on the nanopore morphology [64].

*3.2. Preorganization of the M-Catalyst and Reagents, Mutual Orientation and Shaping the Reaction Start Complex*

The conversion of methane to acetic acid on Fe-ZSM-5 with ultrahigh selectivity has been attributed to the preorganization of the M-catalyst and reagents, the direct coupling of intermediate methyl radicals (●CH$_3$) and the adsorbed CO* and OH* species on Fe site to form CH$_3$COOH (Figure 1c) [72]. Combining NMR analyses and molecular modelling showed significant differences in shape between the different complexes derived from α-, β- or γ-CD (Figure 7) [73].

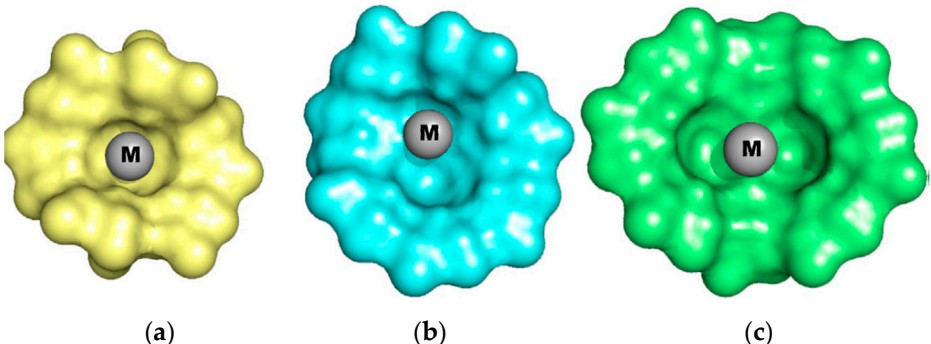

(a)　　　　　　　　　(b)　　　　　　　　　(c)

**Figure 7.** 3D structures of hybrid cyclodextrin–imidazolium M-cavitins: (**a**) α-ICD, (**b**) β-ICD and (**c**) γ-ICD. Reprinted with permission from (Roland et al., 2018) [74]. Copyright 2018 Wiley.

In the case of α-ICD a helical shape is apparent, when in the case of γ-ICD a symmetrical cavity shape is revealed [74]. The preorganization of a substrate in a higher energy conformation can accelerate the reaction and promote reactivity. Nanoenvironments allow (or enforce) the preorganization of substrates through conformational restrictions. Increasing the local concentrations, the mutual convergence and orientation of the catalyst and substrate enables the formation of the start reaction complex. For example, self-assembled nanospheres bearing guanidinium binding sites (Figure 8) bind sulfonate-functionalized ruthenium catalysts increasing the proximity of incoming water to the catalyst. This preorganization increases the reaction rate for electrochemical water oxidation in two-orders of magnitude comparable to the homogeneous system.

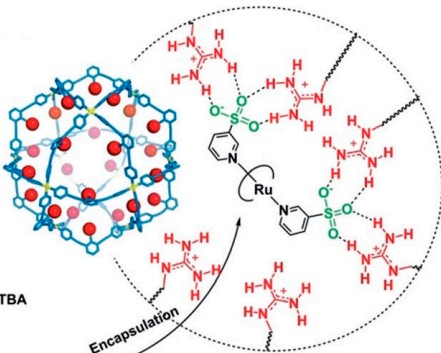

**Figure 8.** Ru-guanidinium nanosphere with incorporated ruthenium complex. Adapted with permission from (Yu et al., 2018) [90]. Copyright 2018 Wiley.

*3.3. Transition State and Intermediate Stabilization*

In cavitins the transition state of the target reaction can be stabilized more efficiently in comparison with bulk solutions. The Ir complex was incorporated into Zr-MOC-NH$_2$ with the formation Ir$^{III}$-MOC-NH$_2$. DFT calculations, mass spectrometry and in situ IR showed that the Ir(III) complex is the catalytic center, and −NH$_2$ in the cavity plays a synergistic role in the stabilization of the transition state and Ir·CO$_2$ intermediate [75]. The transition state or intermediate stabilization can not only lower the energy and enthalpy barrier of the reaction but can also alter the reaction mechanisms. The increased local concentration of reagents in the hydrophobic cage in the cobalt-catalyzed cyclopropanation of styrene, which involves radical intermediates and some shielding, reduces the number of unwanted side reactions of reactive radical intermediates and substantially improves performance [76]. The hexameric resorcinarene capsule (Figure 3a) can host trityl carbocation, which catalyzes Diels–Alder reactions between dienes and unsaturated aldehydes. The capsule promotes the formation of trityl carbocation from trityl chloride via the cleavage of the C-X bond promoted by OH/X H-bonding [77]. The labile imine and hemiaminal intermediates in the transformation of aldehydes to imines can be stabilized in water by hydrophobic cavitin containing a primary amine groups anchored in its cavity (Figure 9). The reaction favors the release of water from the hydrophobic microenvironment [78].

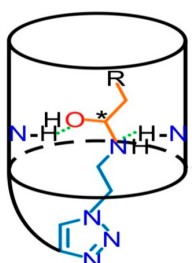

**Figure 9.** Stabilization of labile imine and hemiaminal intermediates through hydrogen bonding and hydrophobic effects by an endo-functionalized cavitin (Li et al., 2022) [78].

### 3.4. Second Sphere and Allosteric Effects

The interfacial interactions between substrate molecules and the cavity, such as the second sphere effect, can not only modulate the first sphere effect but also change the mass transport and adsorption–desorption equilibrium, thus significantly influencing the catalytic reaction activities and selectivities. For example, the electrostatic effects lower the activation energy of a reaction, and result in a rather large rate acceleration. Controlling the reactivity with the presence of acid–base residues and H-bonds is also very significant [20]. Among the eight water molecules embedded in the structure (Figure 3a), four feature a hydrogen atom hanging inside the cavity, which makes them good hydrogen-bond donors to azido ligands of $[Cu(TMPA)N_3]ClO_4$ serving as a reference probe of the second coordination sphere using IR spectroscopy. The presence of this hydrogen bonding was confirmed with docking experiments. The allosteric effects of central and peripheral interactions were extensively investigated. The accumulation of ions or neutral molecules at the periphery of a cavitin can results in a higher local concentration of substates inside of cavity. These peripheral interactions may be non-covalent or covalent in nature. The concentration of externally bound species decreases host flexibility and thus the guest exchange rate. The peripheral cage substituents control the activity of a caged cobalt-porphyrin-catalyst in cyclopropanation reactions. It was demonstrated that cage catalysts with non-polar external groups provided a higher activity compared to the free bulk catalyst and cages with no or polar exo-functionalization [76]. In some M-enzymes the substrate must diffuse through tunnel residues before binding to the active site. For example, the structure of cytochrome P450, which consists of a long hydrophobic tunnel, regulates substrate access and product release. The authors [76] declared that the cage serves as a mimic of the active site pocket of an enzyme whereas the periphery of the cage provides a synthetic equivalent of the substrate binding site tunnel. Mechanistic investigations into the role of the secondary coordination sphere and beyond on multi-electron electrocatalytic reactions showed that the introduction of additional interactions through the secondary coordination sphere beyond the active site, such as hydrogen-bonding or electrostatic interactions, also enables faster chemical steps in addition to its effects on the rate-limiting steps, examined earlier [79]. The cavity containing the M-complex confined in an asymmetric environment permits enantioselective catalytic reactions.

### 3.5. Changing the Reaction Course and Mechanism

The confinement can change the course of a reaction. The reactivity of the catalyst $[Re(^{C12}Anth-py_2)(CO)_3Br]$ was modulated by its encapsulation into a COF. The M-cavitin catalyzed either reductive etherification, oxidative esterification, or transfer hydrogenation depending on the local environment in the COF [80]. In conditions of alkyne hydration by NHC–Au complex the product is formed from intramolecular cyclisation induced by the confinement of the metal. Variations in product distribution were observed with (ICyD)AuCl complexes. A gold-catalyzed enyne cycloisomerization with an $\alpha$-ICyD ligand gave a cyclopentenic product, while $\beta$-ICyD led a to six-membered ring. The outcome of the reaction depends on the conformation of the carbenic intermediate (Figure 10): inside the cavity of the $\alpha$-ICyD conformation a was restricted while b fits better into $\beta$-ICyD [74]. Authors of this work have showed also that changing the size and shape of the cavity also changes the mechanism of reaction.

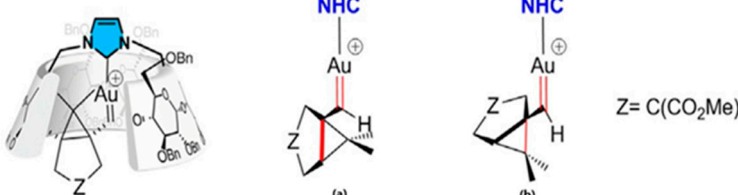

**Figure 10.** Stabilizing the conformation of a reactive intermediate, dependent on CD shape: (**a**) for α-CD and (**b**) for β-CD, determining the product distribution in a gold-catalyzed enyne cycloisomerization. Reprinted with permission from (Roland et al., 2018) [74]. Copyright 2018 Wiley.

*3.6. Regioselectivity, Stereoselectivity and Product Selectivity*

The chemical reaction rate and product selectivity are often dependent on the morphological properties of the cavity in both enzymes and cavitins. The size and shape of the cavity determines the selectivity. The use of ICyD ligands in the copper-catalyzed hydroboration of alkynes leads to an inversion of the regioselectivity that is controlled by a parallel or anti-parallel approach of the alkyne dependent on CD. While the β-ICyD ligand gives branched products, the smaller α-ICyD ligand gives linear vinyl boronates. A chiral second coordination sphere of the M-complex, confined in an asymmetric environment, permits enantioselective catalytic reactions. Such a scenario was demonstrated in enyne cycloisomerization for NHC-capped CD gold complexes (ICyD)AuCl encapsulated in CD in monomeric form. The stereoselectivity depends on the nature of the cyclodextrin and shows good yields and ee values (up to 80%). The selectivities were rationalized using the shapes of the cavities determined by NMR and modelling. While γ-ICyD does not afford enantioselecivity because of its symmetrical shape, α-ICyD and β-ICyD give the enantiomer for which the approach is the easiest according to their helical shape [73]. COFs are capable of inducing chiral molecular catalysts via preferential secondary interactions between the substrate and the framework that induce enantioselectivities not achievable in homogeneous systems. They catalyze the asymmetric acetalization of aromatic aldehydes and 2-aminobenzamide to generate products with up to 93% yield and 97% ee [24]. The highly selective synthesis of terpene compounds was demonstrated by the controlled conversion of (+)-limonene to terpinolene by kinetic suppression of overisomerization in a confined space of a porous metal–macrocycle framework in stark contrast to the acid-catalyzed reaction in bulk solution, which generally gives a complex mixture of thermodynamically favored isomers [81]. The microenvironment of the $\{Co_4^{II}O_4\}$ in the $Co_4@Ru_x$-Eu-MOF plays an important role in improving the performance and selectivity of $CO_2$ photoreduction and water splitting in syngas production. The $H_2$ and CO total yield can be improved by up to 2500 $\mu mol \cdot g^{-1}$ with the ratio of CO:$H_2$ ranging from 1:1 to 1:2 via changing the photosensitizer content in the confined space [82].

Thus, confinement of a metal complex is a promising way to induce their reactivity modulation and improve selectivity. Many of these effects are familiar from enzymology but need be studied in chemical models in order to provide deeper insight into the mechanisms of enzymatic efficiency and selectivity.

**4. M-Cavitins as Advanced Chemical Models of Enzymes**

Although considerable progress in study of enzyme catalysis has been realized by directly monitoring the catalytic processes of natural enzymes, the relationships between the supramolecular structures and the functionality of enzymes are still obscure. Based on their dynamic nature, supramolecular enzyme models with complex and hierarchical architectures have attracted considerable attention in the research area of mimicking the particular features of natural enzymes [83]. The design and development of enzyme mimics with supramolecular structures can help unravel the mystery features of enzyme catalysis. The earlier attempts to adequately model enzyme AC included metal complexes covalently attached to cyclodextrins [84]. Later metal complexes of porphyrin, salen and

others were encapsulated within molecular cages [53]. In an effort to mimic the structure and functions of metalloenzymes discrete coordination and covalent metallocavitins were designed and synthesized either by embedding the active sites in the structures of the cage [85] or by the encapsulation of catalysts within the cage [53]. Being comparable to the AS of enzymes, monocavitins have intrinsic advantages as enzyme mimics. Their solubility in organic solvents permits their study in homogeneous systems [53]. This also facilitates the growing of single crystals that make it possible to reliably control their structure and functionality at the atomic level by using single-crystal X-ray diffraction (SCXRD) [30]. Many spectroscopic techniques, such as NMR, UV–Vis, and fluorescence spectroscopy, have been widely applied to monitor reactions catalyzed by monocavitins.

### 4.1. Stereoselectivity

Stereoselective binding is an essential feature for enzymatic catalysis. A variety of chiral cavitins have been constructed via covalent bonding or coordination assembly [86]. Over the past years, C3 symmetrical cages have emerged as an interesting class of supramolecular hosts that have been reported as efficient scaffolds for chirality dynamics: generation, control, and transfer [87]. Artificial metalloenzymes having a synthetic metal complex in its protein scaffold, selectively catalyze non-natural reactions and reactions inspired by nature in water under mild conditions. For example, the biotin-binding cavity of streptavidin can accommodate small coordination compounds to form the artificial enzyme for the enantioselective oxidation of prochiral sulfides, enhancing the activity and selectivity up to 93% ee for the sulfoxidation of methyl-2-naphthylsulfide in the presence of TBHP compared to the protein-free salt (Figure 11) [13]. Also the highest activity (82% ee with TON 2613) for the enantioselective dihydroxylation and epoxidation of styrene derivatives was obtained using a Ru complex linked with bovine serum albumin [62].

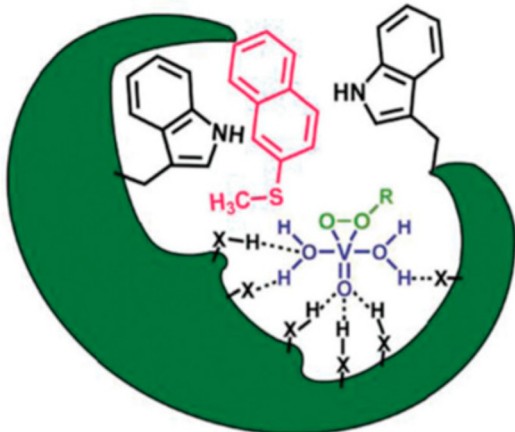

**Figure 11.** Vanadium-dependent artificial peroxidase for enantioselective sulfoxidation reactions. Reprinted with permission from (Dong et al., 2012) [13]. Copyright 2012 Royal Society of Chemistry.

### 4.2. Artificial Photosynthesis

Artificial photosynthesis, including photo-induced water oxidation and $CO_2$ reduction, has been widely studied in the past few years. Native photosynthesis involves three stages: light harvesting, charge separation and redox catalysis, and has special preorganization of chromophores and catalysts. Nature has evolved highly complex and well-organized supramolecular architectures, which can capture sunlight, and transform the solar energy with high efficiency. Cavitins are very suitable for modeling this process as follows from the previous sections. They provide a unique platform designing catalysts for photo-to-chemical energy conversion. Supramolecular chemistry is a powerful tool to achieve larger, more organized molecular structures with an increased level of complexity to optimize properties required for artificial photosynthesis. Like natural systems, perfect preorganization in cavitins leads to improved energy transfer processes,

charge separation and redox catalysis. Porphyrins are synthetically accessible and stable analogues of nature's chlorophylls and therefore have been thoroughly studied as chromophores. Because of their excellent visible light harvesting ability and high electron transfer efficiency, ruthenium bipyridyl complexes are also classic photosensitizers. Thus, the active water oxidation catalyst *cis*-[Ru(bpy)(5,5'-dcbpy)(H$_2$O)$_2$]$^{2+}$ was incorporated into UIO-67 MOF using post-synthetic modification of the framework [88]. XAS, EPR, and Raman spectroscopy confirmed the formation of a highly active Ru$^V$=O key intermediate in M-cavitin [78]. Recently, MOC containing dinuclear and mononuclear Co active sites as well as a [Ru(bpy)$_3$]Cl$_2$ photo-sensitizer and a Na$_2$S$_2$O$_8$ electron scavenger was studied in photo-driven water oxidation (Figure 12) [89].

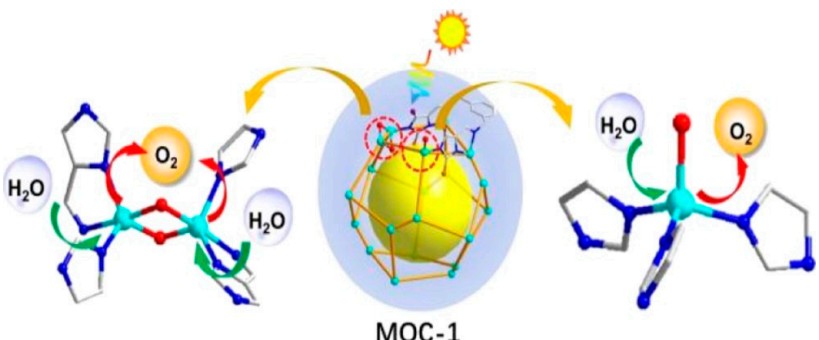

**Figure 12.** A Co MOC for photo-driven water oxidation. Reprinted with permission from (Chen et al., 2021) [89]. Copyright 2021 American Chemical Society.

This study revealed that photo-induced water oxidation initializes the electron transfer from the excited [Ru(bpy)$_3$]$^{2+*}$ to Na$_2$S$_2$O$_8$, and then, the bis(μ-oxo)dicobalt active sites further donate electrons to the oxidized [Ru(bpy)$_3$]$^{3+}$ to drive water oxidation [89]. Self-assembled nanospheres bearing guanidinium-binding sites can strongly bind sulfonate-functionalized ruthenium catalysts. Compared to the homogeneous system, the reaction rate for electrochemical water oxidation was enhanced by two-orders of magnitude by the preorganization of the ruthenium catalysts (Figure 8) [90]. There are more recent examples in Table 1. The study of artificial photosynthesis has great significance for future sustainable development, taking into account converting solar energy into chemical energy, including the production of H$_2$, water oxidation, carbon dioxide and nitrogen fixation, and fine organic syntheses (see next chapters).

**Table 1.** Photosynthesis mimics.

| # | MC | Reaction + hν | Productivity, μmol g$^{-1}$ (Time, h$^{-1}$) | Select. % | Rate, μmol g$^{-1}$h$^{-1}$ (TOF, h$^{-1}$) | References |
|---|---|---|---|---|---|---|
| 1 | FDH@Rh-NU-1000 | CO$_2$ + 2e2H$^+$ → HCOOH | 144 M (24) | nd | (865) | [91] |
| 2 | UiO67-Ir-Cou6 | CO$_2$ → H$_2$ → HCOOH | 26,845 4808 | 95.5 | nd | [92] |
| 3 | MIL-125-Py-Rh | CO$_2$ → HCOOH | 9.5 mM (24) | nd | nd | [93] |
| 4 | CUST-804 | CO$_2$ → CO | nd | 82.8 | 2,710 | [94] |
| 5 | Rh-MOP | CO$_2$ → HCOOH | 76 | nd | (60) | [95] |
| 6 | MAF-34-CoRu | CO$_2$ + H$_2$O → CO | nd | nd | 11.2 | [96] |
| 7 | PFC-58-30 | CO$_2$ + H$_2$O → HCOOH | nd | nd | 29.8 | [97] |

**Table 1.** *Cont.*

| # | MC | Reaction + hν | Productivity, μmol $g^{-1}$ (Time, $h^{-1}$) | Select. % | Rate, μmol $g^{-1}h^{-1}$ (TOF, $h^{-1}$) | References |
|---|---|---|---|---|---|---|
| 8 | POMs@INEP-20 | $CO_2 \rightarrow CO$ | nd | 97.1 | 970 (2.43) | [98] |
| 9 | CABB@M-Ti | $CO_2 \rightarrow CH_4$ | nd | 88.7 | 32.9 | [99] |
| 10 | CTF-Bpy-Co | $CO_2 \rightarrow CO$ | 120 μmol (10) | 83.8 | nd | [100] |
| 11 | SAS/Tr-COF | $CO_2 \rightarrow CO$ | | 96.4 | 980.3 | [101] |
| 12 | RuCOF-TPB | $H_2O \rightarrow H_2$ | 160 | nd | 20,308 | [102] |
| 13 | CoP@ZnIn$_2$S$_4$ | $H_2O \rightarrow H_2$ | nd | nd | 103 | [103] |
| 14 | ZIF-67/CdS HS | $H_2O \rightarrow H_2$ | nd | nd | 1721 | [104] |
| 15 | Ni-Py-COF | $H_2O \rightarrow H_2$ | nd | nd | 626 | [105] |
| 16 | Co-Tz | $H_2O \rightarrow H_2$ | 9,320 | nd | 2,330 | [106] |
| 17 | PTC-318 | $H_2O \rightarrow H_2$ | 80 | nd | nd | [107] |
| 18 | Ru(Bda)-COF | $H_2O \rightarrow O_2$ | nd | nd | 26,000 | [108] |

### 4.3. Models of Redox-Active Enzymes

Hydrogenase enzymes are highly efficient in reducing protons to hydrogen. Both major classes of hydrogenases, the [FeFe]- and [NiFe]-$H_2$ases, contain a bisthiolate-bridged dinuclear complex (FeFe or NiFe, respectively) in active site. The charge transport function of the [4Fe4S]-based enzymatic electron transport chain in redox enzymes is separated from the Fe2 catalytic function, but both these sites are linked as illustrated in Figure 1a for the [FeFe]-$H_2$ase. In the PCN-700 MOF mimic of [FeFe]-$H_2$ase the first function is modeled by an organic redox-active naphthalene diimide-based (NDI) linker, while the Fe2 subsite is modeled by a structurally related [FeFe](dcbdt)(CO)$_6$ (dcbdt = 1,4-dicarboxylbenzene-2,3-dithiolate) complex [109]. The two units reside in preorganized positions within the cavity and the NDI-to-Fe2 distance in the PCN-700/NDI/FeFe is nearly identical in [FeFe]-$H_2$ase. The simple encapsulation of a structural and functional model complex of [NiFe]-hydrogenase into the MOF cavities gives the advanced hydrogenase mimic NiFe@PCN-777 (Figure 13) [110].

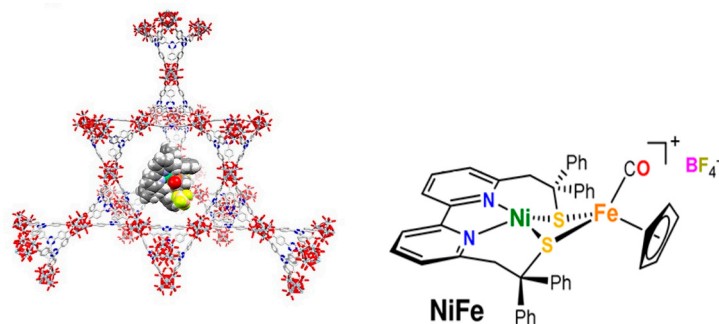

**Figure 13.** Advanced hydrogenase mimic NiFe@PCN-777. Adapted with permission from (Balestri et al., 2017) [110]. Copyright 2017 American Chemical Society.

The preparation and characterization of a redox-active MOF features both a biomimetic model of the hydrogenase active site as well as a redox-active linker that acts as an electron mediator, thereby mimicking the function of [4Fe4S] clusters in the enzyme [109]. MOF-818, containing trinuclear copper centers mimics the active sites of catechol oxidase, possessing efficient catechol oxidase activity with good specificity [111]. Other examples of redox-active enzymes may be found in Table 2.

**Table 2.** Models of redox-active enzymes.

| # | MC | Enzyme | Reaction | Yield, % ($\mu$mol g$^{-1}$) | Select, % | Rate, $\mu$molg$^{-1}$ h$^{-1}$ (TOF, h$^{-1}$) | Refs |
|---|----|--------|----------|------------------------------|-----------|--------------------------------------------------|------|
| 1 | Cu$_3$MOF-818 | Catechol oxidase | DTBC + O$_2$ → DTBQ + H$_2$O$_2$ | 98 | nd | nd | [111] |
| 2 | Cu$_2$MIL-125-Ti | Mono oxygenase | RH + O$_2$ → ROH → epoxide | 94 92 | nd | nd | [112] |
| 3 | Ce-AQ MOF | Mono oxygenase | C$_6$H$_{12}$ +O$_2$ → C$_6$H$_{10}$O | 54.2 | 98.4 | nd | [113] |
| 4 | Ce-UiO-Co | MMO | CH$_4$ +H$_2$O$_2$ → CH$_3$OH + H$_2$O | (2,166,000) | 99 | nd | [114] |
| 5 | Fe/Co-TFT | Hydrogenase | 2H$^+$ + 2e$^-$ ⇌ H$_2$ | nd | nd | (11,000) | [115] |
| 6 | UiO-MOF-Fe$_2$S$_2$ | Hydrogenase | 2H$^+$ + 2e$^-$ ⇌ H$_2$ | 35 | nd | nd | [116] |
| 7 | [Pd$_{12}$(Fe$_2$BB)$_5$ (BBNH$^+$)$_{19}$]$^{43+}$ | Hydrogenase | 2H$^+$ + 2e$^-$ ⇌ H$_2$ | nd | nd | 10,300 mol$^{-1}$s$^{-1}$ | [117] |
| 8 | Mg$_3$(HiTP)$_2$ | Reductase | O$_2$ + 2e,2H+ → H$_2$O$_2$ | nd | 90 | nd | [118] |
| 9 | UiO66(SH)$_2$ | Nitrogenase | N$_2$ + 6e,6H+ → 2NH$_3$ | nd | nd | 32.4 | [119] |
| 10 | NFCO | Nitrogenase | N$_2$ + 6e,6H+ → 2NH$_3$ | 17,600 | nd | 126 | [120] |
| 11 | MIL-101 (Fe$^{II}$/Fe$^{III}$) | Nitrogenase | N$_2$ + 6e,6H+ → 2NH$_3$ | nd | nd | 466.8 | [121] |

*4.4. MMO Mimics*

The development of the direct low-temperature selective oxidation of methane to methanol has remained an active area of research over the last 50 years [122–124]. Native enzymes, the copper-containing particulate methane monooxygenase (pMMO) and the iron-containing soluble methane monooxygenase (sMMO), oxidize methane in ambient conditions by O$_2$ with two connected metal atoms selectively to methanol: CH$_4$ + O$_2$ + 2e$^-$ + 2H$^+$ → CH$_3$OH + H$_2$O (1), NADH → NAD+ + 2e$^-$ + 2H$^+$ (2). The exceptional MMO selectivity is controlled by a special regulatory mechanisms based on the AS hydrophobicity, substrate geometric dimensions, small-molecule tunnel, gate-mechanism and methane quantum tunneling, which together substantially accelerate the oxidation of methane compared to other substrates [68,125–127]. Catalyst encapsulation within a chemically stable porous COF provides a hydrophobic microenvironment around the active site. An anionic macrocyclic catalyst [Fe$^{III}$(Cl)bTAML]$^{2-}$ inside COF nanospheres (Figure 14) permits the oxyfunctionalization of hydrocarbons in water with an enhanced degree of selectivity using the catalyst-immobilized COF nanofilms [128].

It would be interesting to check with this cavitin a fine transformation of some Fe$_2$O$_2$ model complex of the MMO intermediate Q during its interaction with methane or its homologs taking into account following info. The combined structural and theoretical investigation of alkane uptake in a flexible MOF demonstrated accommodation of the C1–C4 alkanes, which are different in size and shape, and reveals that a turn stile mechanism facilitates their transport due to gate-opening [129]. A cavity-tailored MOC containing inward-facing ethyl groups selectively encapsulated methane, ethane, and ethylene at atmospheric pressures in acetonitrile and showed the strongest binding for methane [130]. MOFs bearing Fe(II) sites within Fe$_3$-$\mu_3$-oxo nodes were active for the conversion of CH$_4$ + N$_2$O mixtures via Fe(IV)=O. On the basis of in situ IR spectroscopy and DFT calculations, it was demonstrated that methanol is protected within the MOF under reaction conditions as a methoxy group and its was concluded that there are steps beyond the radical-rebound mechanism to protect the desired CH$_3$OH product [131]. The synthesis of M-cavitin, mod-

eling the coordination environment of the pMMO $Cu_C$ site was reported recently. EPR analysis of the prepared $Cu^{II}$ complex revealed striking similarities to the AS of pMMO. The similar $Cu^{I}$ complex readily reacted with dioxygen and was capable of C-H bond oxidation (Figure 15) [132].

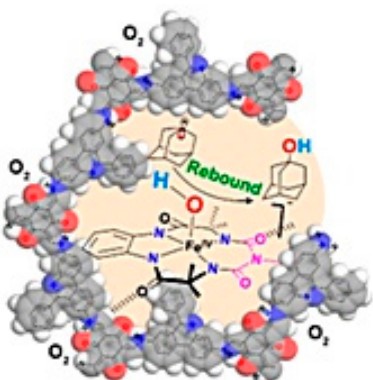

**Figure 14.** C-H functionalization in water via the stabilized $Fe^{IV}=O$ intermediate in the cavity of a COF nanosphere. Adapted with permission from (Sasmal et al., 2021) [128]. Copyright 2021 American Chemical Society.

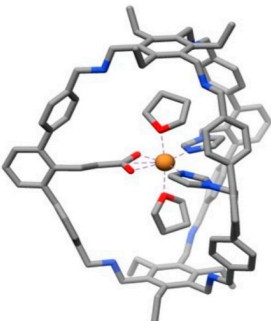

**Figure 15.** Structural and functional mimic of the pMMO $Cu_C$ site. Adapted with permission from (Bete et al., 2022) [132]. Copyright 2022 Wiley.

An artificial binuclear copper monooxygenase $Ti_8$-$Cu_2$ was prepared by metalation of the SBUs in a Ti MOF. The closely spaced $Cu^{I}$ pairs were oxidized by $O_2$ to afford the $Cu^{II}_2(\mu_2$-$OH)_2$ cofactor [112]. The SBU provided a precise binding pocket for the installation of binuclear Cu cofactors to cooperatively activate $O_2$. $Ti_8$-$Cu_2$ showed a turnover frequency at least 17 times higher than that of mononuclear $Ti_8$-$Cu_1$ [112]. Upon incorporation of mononuclear $Fe^{II}$ tris(2-pyridylmetylamine) (FeTPA) into hemicriptophane, a Fe-cavitin was obtained which, in contrast to free $Fe^{II}$TPA, was able to oxidize methane by hydrogen peroxide under 60 °C and 30 bar to methanol [133]. The incorporation of the unstable $Cu^{I}_3L$ complex into mesoporous silica gel allows one to obtain a catalyst which selectively transforms methane to methanol by hydrogen peroxide under room temperature with a conversion of 17.4% and TON 170 (Figure 16) [134].

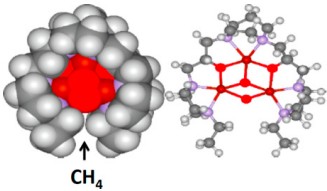

**Figure 16.** The $Cu^{I}_3L$ complex in mesoporous silica gel for the transformation of methane to methanol using hydrogen peroxide under room temperature (Liu et al., 2016) [134].

The authors suggest that this effect is associated with the encapsulation of the complex into the hydrophobic cavity of the silica gel. Unfortunately, the paper does not contain enough evidence to support this suggestion. Nevertheless, the results are very interesting and very much encourage the continuation of this research despite the mechanism not being proven.

Thus, artificial enzymatic systems have been studying to mimic the structures and functions of their natural counterparts. However, there remains a significant gap between modeling and catalytic activity in these artificial systems. Unfortunately, the unlimited possibilities of organic chemistry and supramolecular chemistry have not yet been fully utilized for the biomimetic study of the complex structure of AS MMO and the mechanism of its functioning. It would be very useful to model the conformational effects of the polypeptide scaffold, dynamic changes in the coordination environment of the metal complex, and the sequential formation of intermediates in the multistage process of oxygen and methane activation. Of great interest in this connection is the reproduction of the MMOs catalytic cycle on the base of cavitins and, especially, a deeper penetration into the fine structure of the active intermediate Q and the peculiarities of its transformation during the interaction with methane, for example, using more adequate MMO model complexes [124,135] and advanced enzyme models of types demonstrated in Figures 11–15.

## 5. M-Cavitins in Fine Organic Synthesis

The primary field of MC application is fine organic synthesis and enantioselective catalysis.

Cavitins can induce enantioselectivities not achievable in homogeneous systems involving preferential secondary interactions with the included substrate. For example, they catalyze the asymmetric acetalization of aromatic aldehydes and 2-aminobenzamide with product yields up to 93% at 97% ee [24]. Additionally, the highest activity for the enantioselective dihydroxylation and epoxidation of styrene derivatives was obtained by using a Ru complex linked with a natural cavitin, bovine serum albumin Ru3-BSA-HA [62]. It is well known that special channels in enzymes facilitate the transport of substrates and products. The mesoporous MOF $MnO_2$@OMUiO-66(Ce), containing artificial substrate channels and $MnO_2$ attached to Ce-O clusters, was designed as a super-active artificial catalase [136]. MOF-818, containing trinuclear copper centers that mimic the active sites of catechol oxidase, shows efficient catechol oxidase activity. This artificial enzyme oxidizes o-diphenols to o-quinones with good substrate specificity [101]. The direct selective oxidations of the most difficult C-H bonds with $O_2$ are very challenging reactions and play an important role in fine organic synthesis [137]. Nature has created highly active and selective binuclear metal-containing monooxygenases working in participation with $O_2$ and a reducing agent to activate the most inert C-H bonds of alkanes involving methane. Recently the MOF-based artificial binuclear monooxygenase $Ti_8$-$Cu_2$ was prepared via the metalation of the SBU in a Ti-MOF (see 4.4 [112]). In the presence of coreductants, $Ti_8$-$Cu_2$ demonstrated excellent catalytic activity and selectivity in monooxygenation processes, including epoxidation, hydroxylation and sulfoxidation, with TOF, which is much higher than that of mononuclear $Ti_8$-$Cu_1$ (Figure 17) [112].

It would be interesting to check and develop this M-cavitin for alkane hydroxylation involving methane. While polycavitins are used for the fabrication of advanced heterogeneous catalysts, monocavins are more suitable for the modeling and academic study of enzyme active sites. However, the recent years MOC use in catalysis has also increased. Some examples are shown above. For the case of N-heterocyclic carbene-capped CD gold complexes (ICD)AuCl stereoselectivity in the enyne cycloisomerization depends on the nature of the cyclodextrin: α-ICD and β-ICD give the enantiomer for which the approach is the easiest according to their helical shape and γ-ICD does not afford enantioselecivity because of its symmetrical shape (Figure 10) [74]. The incorporation of iron porphyrin and L- or D-histidine endues chiral COF nanozymes with high activity and selectivity in the peroxidase oxidation of dopa enantiomers. This artificial peroxidase possesses 21.7 times

higher activity than natural HRP [136]. The involvement of organic radical reactions in cavitins helps solve some of the problems connected with the high reactivity and reaction diversity of radicals via taming the reactivity, improving the selectivity or inducing new reaction outcomes [138]. Ti(IV)-based M-calixarene nanocage clusters exhibit extraordinary stability in concentrated acid–alkali solutions and can act as a stable photocatalyst for the oxidation of amines to imines [139]. Other interesting examples may be found in Table 3.

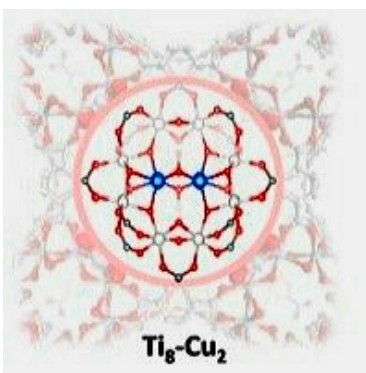

**Figure 17.** The artificial binuclear copper monooxygenase in a Ti MOF, the central circle shows the structure of the Cu bi-nuclear active site. Reprinted with permission from (Feng X. et al., 2021) [112]. Copyright 2021 American Chemical Society.

**Table 3.** Catalysts for fine organic synthesis.

| # | MC | Reaction | Yield, % | Selectivity, % | Rate, mmol $g^{-1}h^{-1}$ (TOF, $h^{-1}$) | References |
|---|---|---|---|---|---|---|
| 1 | PdNPs/ZIF-8 | $PhNO_2 + H_2 \rightarrow PhNH_2$ | 95 | nd | nd | [140] |
| 2 | $(Cu_1Pd_1)PCN$-22(Co) | $2PhY + CO_2 \rightarrow Ph_2CO$ | 90 | 97 | nd | [141] |
| 3 | Br-PMOF(Ir) | $PhEtNH + CO_2 + PhSiH_2 \rightarrow PhEtNCHO$ | 82 | 82 | (507) | [142] |
| 4 | Zn-TACPA | $XC_6H_4NHCHCOOEt + PhCH=CH_2 \rightarrow 3a$ [143] | 91 | nd | nd | [143] |
| 5 | Co@Y | $MeCH=CH_2 + O_2 \rightarrow$ epoxyde | 24.6 | 57 | 4.7 | [144] |
| 6 | CoP@POC | $2PhCH_2NH_2 + O_2 \rightarrow PhCH_2N=CHPh$ | 93 | 99 | (22,989) | [145] |
| 7 | CuHENU-8 | $Me_2PhSiH + t\text{-}BuOOH \rightarrow Me_2PhSiOH$ | 89 | 95 | 132 | [146] |
| 8 | Ni-SAPO-34 | $C_6H_{10}=O + O_2 \rightarrow C_4H_{10}(COOH)_2$ | 30 | 87 | nd | [147] |
| 9 | Prism1 | $ArCH_2OH + O_2 \rightarrow ArCHO$ | 99.9 | nd | nd | [148] |
| 10 | POM/MOF | $PhCH=CH_2 + H_2O_2 \rightarrow PhCHO$ | 96 | 99 | nd | [149] |
| 11 | Zr-abtc | Carvone $+ H_2O_2 \rightarrow$ 1,2-epoxide | 87 | 90 | nd | [150] |
| 12 | SNNU-97-InV | Me-epoxide $+ CO_2 \rightarrow$ Me-$c$-carbonate | 73.3 | 99 | (24.2) | [151] |

**Table 3.** *Cont.*

| # | MC | Reaction | Yield, % | Selectivity, % | Rate, mmol $g^{-1}h^{-1}$ (TOF, $h^{-1}$) | References |
|---|---|---|---|---|---|---|
| 13 | NUC-45a | Ph-epoxide + $CO_2$ → Ph-carbonate | 99 | 99 | (316) | [152] |
| 14 | NUC-54a | PhCHO + $CO_2$ → Ph-carbonate | 98 | nd | (47) | [153] |
| 15 | MOF1 | 1-Et-2Ph-aziridine + $CO_2$ → oxazolidinone | 99 | 97 | nd | [139] |
| 16 | Ent-1(3b) | nerol → α-terpineol +limonen | 73 | 70*ee* | nd | [154] |
| 17 | CuPMO | $NH_2C_6H_4OH + CH_2(COMe)_2$ → benzoazole | 83 | nd | nd | [155] |
| 18 | Ni-Ir@Tp-Bpy | $R^1R^2NH + RJ → R^1R^2NR$ | 94 | nd | nd | [156] |
| 19 | $Cu_2O$@ZIF-8 | $Me_2C(OH)-C≡CH + CO_2$ → α-alkylidenecarbonate | 97 | nd | (3.03) | [157] |
| 20 | (R)-CuTAPBP-COF | $EtCHO+PyCH_2Br$ (R)-MPP | 98 | 95*ee* | nd | [158] |
| 21 | JLU-MOF-112 | $PhCHO + CH_2(CN)_2$ → $PhCH=C(CN)_2$ | 98 | nd | (198) | [159] |
| 22 | Cu-1D MOF | Pyrazole + PhJ → 1-Ph-1*H*-pyrazole | 95 | nd | nd | [160] |
| 23 | PdAg@ZIF-8 | $CH_2=CHC_6H_4NO_2 + H_2$ → $CH_2=CHC_6H_4NH_2$ | 98 | 97.5 | nd | [161] |
| 24 | JNM-4-Ns | $R_1C_6H_4-C≡CR_2 + B_2Pin_2$ → $R_1C_6H_4-CH=CR_2BPin$ | 90 | nd | (41,734) | [162] |
| 25 | UiO-66-$Gua_{0.2}$ | $CO_2 + ECH → CPC$ | nd | nd | (110.3) | [163] |
| 26 | $Bi_2S_3$@quasi-Bi-MOF | $4-NO_2PhOH + H_2$ → $4-NH_2PhOH$ | nd | 97 | nd | [164] |

It is worth to note an excellent reaction rate, high product selectivity and productivity outperforming most reported photocatalysts for MC #6 in Table 3 [145], in which the single Co atom organic cage CoP@POC demonstrates prominent photocatalytic efficiency for the oxidation of amines into imines in visible light.

## 6. M-Cavitins as Promising Industrial Catalysts

The growth of energy consumption and environmental problems have resulted in the search of catalysts for industrial energetically challenging processes with participation of small gas molecules involving innovative reactions, high selective to valuable products. The development of renewable and efficient energy conversion technologies is becoming extremely necessary. These technologies must be based on the principles of biomimetic chemistry and M-cavitin catalysis. For the realization of artificial photosynthesis, it is necessary to develop the design of fast and durable water oxidation catalysts that can be incorporated into future sunlight-to-chemical-fuel assemblies. The activation and transformation of small molecules, such as $CO_2$, $N_2$, $O_2$, $CH_4$ and $H_2$, into other products has always been central to endeavors of chemical science. Among various types of energy conversion, electrochemical $CO_2$ reduction ($CO_2R$) and water splitting (WS) have also been proven as promising strategies for their environmental benignancy and high efficiency. For the small molecules discussed here, the spatial and temporal control of protons and electrons delivery to/from the active site is crucial in maintaining product selectivity in these transformations [165]. The modern challenges of climate change, energy sustainability, and resource efficiency make the activation of small molecules more important than ever before.

### 6.1. H₂O

Water oxidation catalysis is of pivotal importance to progress the field of artificial photosynthesis. Water is an important renewable energy source and has the potential to meet the current energy crisis needs via photochemical, electrochemical, and photoelectrochemical splitting to produce oxygen and hydrogen green fuels. Water splitting is comprised of two half-cell reactions: an oxygen evolution reaction (OER) and a hydrogen evolution reaction (HER). The facile synthesis and electrocatalytic HER performance of SnTPPCOP was demonstrated recently (Figure 18) (Table 4, #16), which exhibited good HER activity with a low overpotential of 147 mV at 10 mA cm$^{-2}$ due to its unique structural properties, ranking among the best new reports.

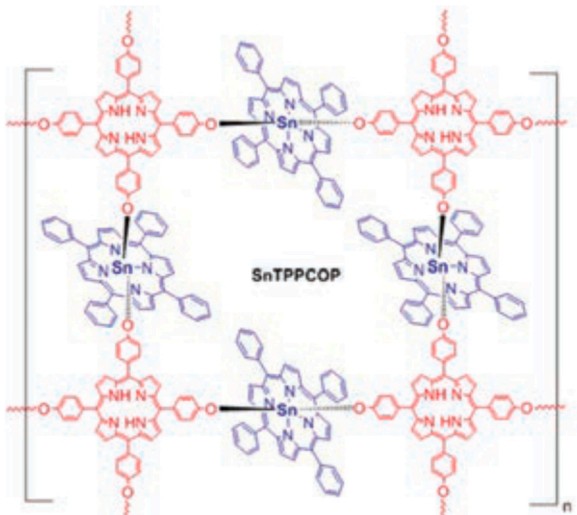

**Figure 18.** Tin porphyrin axially-coordinated 2D covalent organic polymer (SnTPPCOP) for HER activity (Wang Q. et al., 2022) [179].

**Table 4.** Activation of small molecules for sustainable development.

| # | MC | Reaction | Yield, % (P, µmol g$^{-1}$/FE, %) | Selectivity, % | Rate, µmol g$^{-1}$h$^{-1}$ (TOF h$^{-1}$) | OP, mV (CD, mA sm$^{-2}$) | Refs |
|---|---|---|---|---|---|---|---|
| 1 | Fe/Ni-MOF | $H_2O \rightarrow O_2$ | nd | nd | (940) | 239 (50) | [166] |
| 2 | CALF20 | $CO_2 \rightarrow CO$ | (nd/94) | nd | (1361) | (32.8) | [167] |
| 3 | HNTM-Ir/Pt | $H_2O \rightarrow H_2$ | (600) | nd | 201.9 | nd | [168] |
| 4 | Ir$^{III}$-Uio-67-NH$_2$ | $CO_2 \rightarrow CO$ | nd (nd/6.71) | 99.5 | (120) | nd | [75] |
| 5 | TiO$_2$@ZIF-8 | $H_2O \rightarrow H_2$ | 51 [1] | nd | 262,000 | nd | [18] |
| 6 | (NiCo)S$_2$/NCNF | $H_2O \rightarrow O_2 \rightarrow H_2$ | nd | nd | nd | 177 (10) 203 (10) | [169] |
| 7 | ZnO/Fe$_2$O$_3$ PN | $CH_4 \rightarrow CH_3OH$ | (178.3/nd) | 100 | nd | nd | [170] |
| 8 | Mn$_1$Co$_1$/CN | $CO_2 \rightarrow CO$ | nd | nd | 47 | nd | [171] |
| 9 | ZPMOF | $CO_2 \rightarrow CH_4$ | nd | 70 | 32 | nd | [172] |
| 10 | T1-2Cu | $CO_2 \rightarrow CH_4$ | nd | 93 | 3.7 | nd | [173] |
| 11 | NiFe-MOF/FF | $H_2O \rightarrow O_2$ | nd | 83.8 | nd | 216 (50) | [174] |
| 12 | Cu@FCN MOF/CF | $H_2O \rightarrow O_2$ | nd | 88.7 | nd | 290 (10) | [175] |

**Table 4.** *Cont.*

| # | MC | Reaction | Yield, % (P, $\mu mol\ g^{-1}$/FE, %) | Selectivity, % | Rate, $\mu mol\ g^{-1}h^{-1}$ (TOF $h^{-1}$) | OP, mV (CD, mA sm$^{-2}$) | Refs |
|---|---|---|---|---|---|---|---|
| 13 | Fe$_3$-MOF-BDC-NH$_2$ | $H_2O \rightarrow O_2$ | nd | nd | nd | 280 (10) | [176] |
| 14 | CoCu-MOF NBs | $H_2O \rightarrow O_2$ | nd | nd | 1084 | 271 (10) | [177] |
| 15 | CuNi-NKU-101 | $H_2O \rightarrow H_2$ | nd | 100 | nd | 324 (10) | [178] |
| 16 | SnTPPCOP | $H_2O \rightarrow H_2$ | nd | nd | nd | 147 (10) | [179] |
| 17 | CoP/CNTHPs | $H_2O \rightarrow O_2$ $\rightarrow H_2$ | nd | nd | nd | 238 (10) 147 (10) | [180] |
| 18 | Ru/3DMNC | $H_2O \rightarrow O_2$ $\rightarrow H_2$ | nd | nd | nd | 217 (10) 51 (10) | [181] |
| 19 | MnZn-MUM-1/NF | $H_2O \rightarrow O_2$ | nd | nd | (83.3) | 253 (10) | [182] |
| 20 | Co-Fe-P | $H_2O \rightarrow O_2$ | nd | nd | nd | 240 (10) | [183] |
| 21 | Ce-Ni(OH)$_2$ @Ni-MOF | $H_2O \rightarrow O_2$ | nd | nd | (170) | 272 (100) | [184] |
| 22 | FePc-pz | $N_2 \rightarrow NH_3$ | (nd/31.9) | nd | 33.6 [2] | nd | [185] |
| 23 | Fe$_1$S$_x$@TiO$_2$ | $N_2 \rightarrow NH_3$ | (nd/17.3) | nd | 18.3 [2] | nd | [186] |
| 24 | UiO-66-H | $CH_4 \rightarrow CH_3OOH$ | nd | 100 | 350 | nd | [187] |

[1] quant eff; [2] $\mu g\ \mu g\ h^{-1}$.

Photo-induced WS into hydrogen and oxygen has been perceived as one of the most promising pathways for solving the energy crisis and environmental problems. The double-shelled TiO$_2$@ZIF-8 hollow spheres used for HER under illumination show efficient charge separation by electron injection from ZIF-8 to TiO$_2$, high photocatalytic quantum efficiency and a high HER rate, 3.5 times higher than bare TiO$_2$ (Table 4, #5) [18]. Donor–acceptor type imine-linked COFs can be produced, under visible light irradiation, upon protonation of their imine linkages. A significant redshift in light absorbance, largely improved charge separation efficiency, and an increase in hydrophilicity triggered by protonation of the Schiff-base moieties in the imine-linked COF are responsible for the improved photocatalytic performance [188]. Electrocatalytic WS has been regarded as one of the most promising approaches for producing hydrogen under mild conditions. Despite many progresses achieved in electrocatalytic WS, highly active and durable catalysts have to be developed to overcome the kinetic barriers in the water splitting process, especially for the OERs [189]. The theoretical basis for the design of new MOF electrocatalysts was recently elaborated through a study of the relationship between the structure and properties of trimetallic MOFs for efficient OERs. Fe$_3$-MOF-BDC-NH$_2$ exhibited an enhanced performance, superior to other reported catalysts (Table 4, #13) [176]. The multi-shelled hollow Mn/Fe-hexaiminobenzene MOF (Mn/Fe-HIB-MOF), featuring a conductive skeleton, was developed as an excellent bifunctional electrocatalyst for oxygen reduction reactions and OERs. It exhibited high OER performance, outperforming commercial RuO$_2$, Mn-HIB-MOF and Fe-HIB-MOF catalysts [190]. The FeNi-MOF showed remarkable electrocatalytic performance with a low overpotential of 266 mV at 100 mA cm$^{-2}$, and a high TOF value of 0.261 s$^{-1}$ at an overpotential of 270 mV as well as superb long-term durability with a high current tolerance for water oxidation [166] (Figure 19).

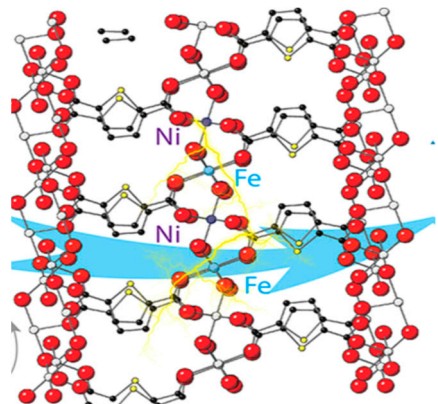

**Figure 19.** FeNi-MOF electrocatalyst for water oxidation. Reprinted with permission from (Wang CP et al., 2021) [166]. Copyright 2021 American Chemical Society.

Photo-induced water oxidation by an MOF has been widely studied in the past few years. The active water oxidation catalyst *cis*-[Ru(bpy)(5,5′-dcbpy)(H$_2$O)$_2$]$^{2+}$ was incorporated into a UIO-67 MOF using post-synthetic modification of the framework. OER was studied using an oxygraph with a Clark electrode at pH = 1. XAS, EPR, and Raman spectroscopy confirmed the formation of M-cavitin and the highly active Ru$^V$=O key intermediate [88]. MOCs based on cobalt ions and imidazolate ligands were studied on water photo-oxidation for the first time. These studies revealed that the reactions initialized via the electron transfer from the excited [Ru(bpy)$_3$]$^{2+*}$ to Na$_2$S$_2$O$_8$, and then, to the bis(μ-oxo)dicobalt active sites which further donated electrons to the oxidized [Ru(bpy)$_3$]$_3$$^+$ to drive water oxidation [89]. Recent advances in research of MOF nanoarchitectures for efficient electrochemical water splitting have reviewed [191]. Hierarchical bifunctional catalysts for WS are the most promising catalysts for energy transformation in future. For example, the bifunctional catalyst CoP/CNTHP, containing non-precious metals for efficient water splitting, has been shown to have outstanding catalytic activity and stability for overall WS [180].

*6.2. CO$_2$*

Because of the highly stacked layers, some COFs adopt semiconductive properties and exhibit promising catalytic performances in photo-CO$_2$R [192]. 3D flower-like SnS$_2$ with a sheet structure shows good performances for efficient CO$_2$ photoreduction under visible-light irradiation [193]. Under visible-light irradiation, the single Ir$^{III}$-MOC-NH$_2$ cage can convert CO$_2$ into CO with high selectivity and a TOF which is 3.4 times as much as bulk Ir$^{III}$-MOC-NH$_2$ and two orders of magnitude greater than that of the classical MOF counterpart, Ir$^{III}$-Uio-67-NH$_2$ [75] (Table 4, #4). The redox active In-MOF, In$^{III}$[Ni(C$_2$S$_2$(C$_6$H$_4$COO$_2$)$_2$], demonstrates the first example of a Ni-based MOF catalyst in electrocatalytic CO$_2$R, which opens promising prospects for designing novel and efficient non-noble metal-based, redox-active, biomimetic MOFs [194]. Conductive two-dimensional phthalocyanine-based MOF (NiPc-NiO$_4$) nanosheets linked by nickel-catecholate, are highly efficient electrocatalysts for CO$_2$R to CO electroreduction. The obtained NiPc-NiO$_4$ has good conductivity and exhibits a very high selectivity of 98.4% toward CO production and a large CO partial current density of 34.5 mA cm$^{-2}$, outperforming the reported MOF catalysts [195]. The MOF UiO-66 was used in tandem with its zirconium oxide nodes and incorporated ruthenium PNN pincer complex to hydrogenate carbon dioxide to methanol giving the highest reported turnover number (TON) (19,000) and turnover frequency (TOF) (9100 h$^{-1}$). Moreover, the reaction was readily recyclable, leading to a cumulative TON of 100,000 after 10 reaction cycles [196]. The neighboring Zn$^{2+}$-O-Zr$^{4+}$ sites obtained by post-synthetic treatment of Zr$_6$(μ$_3$-O)$_4$(μ$_3$-OH)$_4$ nodes of MOF-808 by ZnEt$_2$ gave the MOF-808-Zn catalyst, which exhibited a >99% MeOH selectivity in CO$_2$ hydrogenation at 250 °C and good stability for at least 100 h [197] (Figure 20).

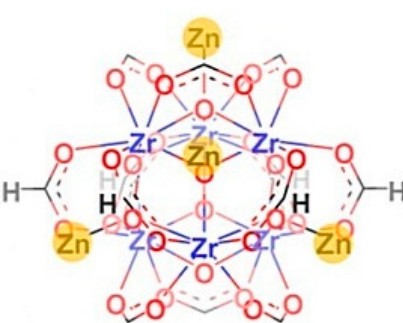

**Figure 20.** Zn-Zr cluster in the MOF-808-Zn catalyst for $CO_2$ hydrogenation (Zhang J. et al., 2021) [197].

Mechanistic investigations revealed that $Zn^{2+}$ is responsible for $H_2$ activation and the $Zn^{2+}-O-Zr^{4+}$ site is critical for $CO_2$ adsorption and conversion.

Along with WS, the electrochemical reduction of carbon dioxide ($CO_2R$) has also been proven to be a promising strategy among various types of energy conversion. A desired electrocatalyst should have a high TON, a high TOF and a small overpotential. In cavitins the transition state of the target reaction can be stabilized more efficiently in comparison with bulk solution. The Ir complex was incorporated into Zr-MOC-NH$_2$ with the formation of Ir$^{III}$-MOC-NH$_2$. DFT calculations, mass spectrometry and in situ IR showed that the Ir(III) complex is the catalytic center, and $-NH_2$ in the cavity plays a synergistic role in the stabilization of the transition state and the Ir·$CO_2$ intermediate (Table 4, #4) [75]. Ni-MOF-derived catalysts for the light-driven methanation of $CO_2$ under UV–Vis IR irradiation displayed excellent recyclability without the loss of catalytic activity [198]. The high load of Zn-porphyrin in an anionic porous Zn-based PMOF has strong kinetic and thermodynamic advantages demonstrating a good performance in the photocatalytic $CO_2$-to-$CH_4$ conversion due to the atomically dispersed active catalytic sites and fast charge transfer (Table 4, #9) [172]. In another example, $Cu^{2+}$ ions were dispersed in the crystal structure of the MOF matrix. The doping content of the $Cu^{2+}$ ions and the photocatalytic performance displayed a volcanic relationship: the medium concentration (1Ti/2Cu) was optimal for the greatest performance for $CH_4$ and CO (3:1) (Table 4, #10) [173]. The d-UiO-66/$MoS_2$ composite facilitates the photo-catalytic conversion of $CO_2$ and $H_2O$ to $CH_3COOH$ under visible light [199]. Multiple Cu centers supported on the Ti-MOF catalyze $CO_2$ hydrogenation to ethylene and presents a new tandem route for $CO_2$-to-$C_2H_4$ conversion via $CO_2$ hydrogenation to ethanol followed by its dehydration [200]. Bifunctional MOFs containing tripyridyl complexes of Fe and Mn convert styrene into styrene carbonate via tandem epoxydation using $O_2$ and then $CO_2$ insertion. DFT calculations revealed the involvement of a high-spin Fe$^{IV}$ (S = 2) center in the challenging oxidation of the sp$^3$ C-H bond [201]. A new porous copper$-$organic framework assembled from 12-nuclear [Cu12] nanocages with two types of nanotubular channels and a large specific surface area effectively catalyzed the cycloaddition of $CO_2$ to various epoxides under mild conditions [202]. Electrocatalytic $N_2$ reduction reactions (NNRs) at ambient conditions is a good way for sustainable $NH_3$ production, because the latter is a valuable raw material in organic synthesis and a significant clean energy carrier. The pyrazine-linked iron-phthalocyanine FePc-pz is an efficient electrocatalyst for simultaneously enhancing NRR activity and selectivity and is the best among the NNR electrocatalysts (Figure 21) (Table 4, #22) [185]. Inspired by the natural nitrogenase, the single-atom M-cavitin containing Fe$_1$S$_x$ in mesoporous $TiO_2$ appears as an excellent catalyst with a high rate and efficiency for NNR ((Table 4, #23) [186].

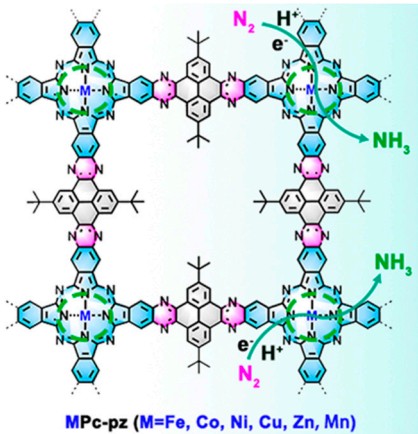

**Figure 21.** Pyrazine-linked iron-phthalocyanine FePc-pz for electrocatalytic $N_2$ reduction (Zhong H. et al., 2021) [185].

*6.3. Methane*

Compared to WS and $CO_2$ conversion, the effective and selective chemical routes to valorize the most abundant hydrocarbon on earth with the participation of cavitins are relatively less studied. Direct methane conversion has been carried out in the gas phase over Cu- and Fe-containing zeolites in the stepwise cyclical process that firstly involves interaction of the transition metal with $O_2$ or $N_2O$ at 400–500 °C, forming an oxidative intermediate, then the methane reaction with this active intermediate at 200 °C, and finally the product extraction with water steam. However, the rate and productivity of these processes are still very low. A catalytic process with zeolite Fe-Cu-ZSM-5 with a selectivity of 20–80% was proposed for the hydroxylation of methane at 50 °C using aqueous $H_2O_2$ [203,204]. Such a Cu-Fe (2/0.1)/ZSM-5 catalyst is an efficient catalyst for the direct conversion of methane into methanol with excellent productivity and a methanol selectivity of 80% [204]. The mechanism based on the catalytic, spectroscopic and theoretical results was suggested: [204] the adjacent to the iron acid sites facilitates the formation of an active Fe(V)=O intermediate via the dehydration of the formed Fe-OOH in aqueous $H_2O_2$ solution, enabling the homolytic cleavage of the primary C-H by a radical-rebound mechanism to generate •$CH_3$ radicals that are quickly captured by •OH radicals to form $CH_3OH$. In contrast to Cu- and Fe-containing zeolites, the study of MOF-based MMO mimics is still in the early stages and suffers from the same problems, low productivity and rate and low methanol selectivity due to over-oxidation. The significant challenge for $CH_4$ photooxidation into $CH_3OH$ are connected to activation of the inert C-H bond and inhibition of $CH_3OH$ over-oxidation. In this connection several interesting works have recently appeared. Thus, it was shown that in reaction $CH_4 + H_2O_2 = CH_3OOH$, catalyzed by UiO-66-H, the high electronic density on Zr-oxo nodes facilitates the formation of Zr-oxo/•OH intermediates which are competent to activate the methane C-H bond with 100% selectivity (Table 4, #24) [187]. It was suggested then that, due to DFT calculations, the Zr-oxo/•OH intermediates can quickly react with the •OOH or the dissolved $O_2$ with an extremely low energy barrier, explaining the formation $CH_3OOH$. The absolute selectivity is a very unusual result for direct methane oxidation, but its explanation is not convincing enough. Indeed, if this intermediate is competent in the reaction with methane [187], why is it not competent with $CH_3OH$ [187]? The electron donor–acceptor hybrid RhB/$TiO_2$ demonstrated the photocatalytic oxidation of $CH_4$ to $CH_3OH$ with a rate 143 µmol·$g^{-1}$·$h^{-1}$ and a selectivity 94% in ambient conditions, utilizing visible light [205]. On one hand, two metal sites with different electronegativities can modulate the activity of $CH_4$ activation and inhibit the overoxidation of $CH_3OH$. (Table 4, #7) [170]. The ZnO/$Fe_2O_3$ porous nanosheets efficiently performed $CH_4$ hydroxylation and suppressed $CH_3OH$ overoxidation through strengthening its O−H bond. The experimental results and DFT calculations confirmed

that the $ZnO/Fe_2O_3$ heterojunction results in a higher charge accumulation at the Fe sites through a charge transfer from the Zn sites, which favors the adsorption of $CH_4$ molecules and further helps to lower the rate-limiting barrier of $CH_3OH$ generation. On the other hand, these Fe sites endow the O-H bond of $CH_3OH$ with a higher polarity through the transferring of electrons to the O atoms, this inhibits the homolytic cleavage of the O-H bond to generate highly reactive radicals [170]. Other examples of small molecule activation may be found in Table 4.

## 7. Conclusions and Outlooks

Cavities and other holes are ubiquitous in the material word. Enzymes are natural cavitins. They have evolved over millions of years to provide extremely powerful catalysts toward a variety of reactions with excellent activities under mild conditions and exquisite substrate specificity and product selectivity. Our fundamental understanding of enzymatic catalysis has inspired scientists to develop and explore smaller synthetic complexes as enzyme mimics. Chemical cavitins have emerged due to the efforts of synthetic and supramolecular chemists. They represent an emerging class of molecules and supramolecular ensembles with intrinsic porosity. In-depth studies of these biomimetic artificial systems have provided important insights into natural enzymes [206]. However, there remains a significant gap between the structural modeling and catalytic activity in these artificial systems [112]. Due to advances in synthetic chemistry a huge diversity of cavitins inspired by enzymes has appeared during the last decade [8]. The ease of their synthesis has already provided us with a rich library of architectures. Their further functionalization to afford multifunctional assemblies is the challenge ahead [207]. Though significant advances have been achieved in recent years in cavitin chemistry [207], greater insights into the many subtle factors affecting their shape and size as well as cavity effects on catalysis are still required [208]. Compared to natural self-assembly, chemical cavitins often lack complexity, a feature highly desirable for enzyme mimics and advanced bioinspired catalysts [54,209,210]. Confined in a cavity, molecules can fundamentally change their chemical and physical properties compared to those in bulk solution [211]. Computational methods, advanced machine learning models, and direct and powerful techniques, such as in situ X-ray diffraction or single-crystal characterization, could be significant to better understanding cavity effects [212]. As demonstrated in recent studies, operando XAFS and FTIR techniques have been used as powerful tools for monitoring the evolution of the reactive centers at the molecular level. Although tremendous attention has been devoted to the development of single-atom catalysts [208], only a few reports are related to the construction of dinuclear and multi-nuclear metal species in cavitins. The exciting results were achieved during the last decade in the single-electron redox reactions [213]. At present, the greatest challenge to multielectron and proton photochemical transformations has appeared [214]. M-cavitins have opened up the possibility for chemistry to use the many principles developed in nature during evolution, expanded the tools of organic chemistry and have proven extremely important in fine organic synthesis and pharmaceutical chemistry, especially for enantioselective reactions. They not only improved the efficiency and selectivity of a number of reactions, but also allowed them to change their direction to new products. M-cavitins demonstrate high activities for energetically challenging reactions with the participation of small gas molecules and high selectivity to valuable products [215]. Great achievements have already been made in clean photocatalytic and electrochemical energy conversion using affordable and inexhaustible clean materials, such as water and carbon dioxide, which can produce valuable fuels and chemicals [216]. Using ubiquitous visible-light irradiation to reduce $CO_2$ to C-based products is an environmental and economic method which transforms solar energy in the form of chemical bonds. Looking forward, innovation efforts are still necessary to help solve the global energy crisis [217–219]. Achievements and solutions to this problem will be connected primarily with the development of fundamental scientific research in bioinspired catalysts [207]. These studies will require completely new approaches for the design and synthesis of a more diverse library of M-cavitins with

innovative structures, metal ion composition and functionality. Photo- and electrocatalysis of other abundant resources, such as $N_2$ and methane, are now coming increasingly into focus. In particular, efforts to decipher the reaction mechanisms and extract fundamental insights are necessary to develop economically competitive routes using the direct methane oxidation to methanol [220]. The direct oxidation of methane in a laboratory setup with the participation of M-cavitins using mild conditions is still a challenging problem [221]. There is a huge gap between MMO and chemical catalysts based on M-cavitins, not only in activity and selectivity but also in the mechanism of direct oxidation of methane to methanol [220]. Though there still remains a considerable gap between the academic research and industrial applications [222], the numerous works covered in this review demonstrate a promising foundation for the future.

**Funding:** This research received no external funding.

**Data Availability Statement:** No new data were created or analyzed in this study. Data sharing is not applicable to this article.

**Acknowledgments:** The author would like to thank the Russian Ministry of Science and Education for supporting this work (via project no. AAAA-A19-119071190045-0 to the IPCP RAN), and Vera Kosareva for her help in the design of the figures.

**Conflicts of Interest:** The author declare no conflict of interest.

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
