# Peer review of "Metallocavitins as Advanced Enzyme Mimics and Promising Chemical Catalysts"

_catalysts, doi:10.3390/catal13020415_

Round 1

Reviewer 1 Report

It is known to all that the cavity structure of molecular nanocontainers and the real possibility of modifying their cavities provide unlimited possibilities for simulating the active centers of metalloenzymes. In this manuscript, the review focuses on figuring out how chemical reactivity is controlled in a well-defined cavitin nanospace and summarize them. in addition, the author discusses advanced metal-cavitin catalysts related to the study of the main stages of artificial photosynthesis, as well as to highlight the current challenges of activating small molecules. On the whole, the framework for the review is reasonable, and the content is substantial. I consider that this review is suitable for publication on Catalysts. However, there are a few points which the authors should adjust before publication.

(1)   The definition of some figures is low. It is recommended to provide high-quality figures, such as Figure 2d, Figure 16, Figure 18, Figure 19. At the same time, the corresponding literature source and publishing group should be given below the Figures.

(2)   It is suggested to provide some reaction mechanism diagrams, not just structural formulas.

(3)   Some important literatures on photocatalytic water splitting and CO2 reduction have not been paid attention to, and the following important literature about are encouraged to be cited: OER (Angew. Chem. Int. Ed. 2020, 59, 19691-19695), CO2RR (Inorg. Chem., 2021, 60, 18598-18602)

Author Response

First of all, I thank the reviewer for helpful comments.

Low-quality figures have been replaced or their quality has been improved. The captions under the figures are expanded. Recommended links are included. Regarding the proposal to give schemes of mechanisms: the mechanisms of individual reactions in the absence of cavitin depend on the electronic nature of the metal, the structure of the metal complex and the medium and are not the subject of my review. The general mechanism of the catalytic action of cavitins is discussed in the sections Introduction, Cavity Effects and at the discussion of each catalyst.

Reviewer 2 Report

Recommendation: Accept with minor revision

Comments:
The claim of the review is "Metallocavitins as Advanced Enzyme Mimics and Promising

Chemical Catalysts" by Albert A. Shteinman, which is a well-chosen topic. I would like to recommend this work to ‘Catalysts’ after the following minor correction.

Give the general mechanism of catalyst for the chemical transformation in the introduction section.

Author Response

First of all, I thank the reviewer for helpful comment.

The general mechanism is given in Figure 1b.

Reviewer 3 Report

In this review, Shteinman focus on figuring out how chemical reactivity is controlled in a well-defined cavitin nanospace. The author also intends to discuss advanced metal-cavitin catalysts related to the study of the main stages of artificial photosynthesis, including energy transfer and storage, water oxidation and proton reduction, as well as to highlight the current challenges of activating small molecules such as H2O, CO2, N2, O2, H2, CH4. The supramolecular approach is becoming increasingly dominant in biomimetics and chemical catalysis due to the expansion of the enzyme active center idea, which now includes binding cavities, channels and canals for transporting substrates and products. 

this review is well organized and well written, this referee is sure that this review has broad audience in supramolecular chemistry., catalysis, and also process Chemistry. therefore this referee strongly recommend it to be published on the Journal ”Catalysts” in this present state. the typo might be corrected during proof-reading process. 

Author Response

There were no comments. I thank the reviewer for nice evaluation of my work.